# Deep Learning-Based Reconstruction for Cardiac MRI: A Review

**DOI:** 10.3390/bioengineering10030334

**Published:** 2023-03-06

**Authors:** Julio A. Oscanoa, Matthew J. Middione, Cagan Alkan, Mahmut Yurt, Michael Loecher, Shreyas S. Vasanawala, Daniel B. Ennis

**Affiliations:** 1Department of Bioengineering, Stanford University, Stanford, CA 94305, USA; 2Department of Radiology, Stanford University, Stanford, CA 94305, USA; 3Department of Electrical Engineering, Stanford University, Stanford, CA 94305, USA

**Keywords:** cardiac magnetic resonance imaging, image reconstruction, machine learning, deep learning

## Abstract

Cardiac magnetic resonance (CMR) is an essential clinical tool for the assessment of cardiovascular disease. Deep learning (DL) has recently revolutionized the field through image reconstruction techniques that allow unprecedented data undersampling rates. These fast acquisitions have the potential to considerably impact the diagnosis and treatment of cardiovascular disease. Herein, we provide a comprehensive review of DL-based reconstruction methods for CMR. We place special emphasis on state-of-the-art unrolled networks, which are heavily based on a conventional image reconstruction framework. We review the main DL-based methods and connect them to the relevant conventional reconstruction theory. Next, we review several methods developed to tackle specific challenges that arise from the characteristics of CMR data. Then, we focus on DL-based methods developed for specific CMR applications, including flow imaging, late gadolinium enhancement, and quantitative tissue characterization. Finally, we discuss the pitfalls and future outlook of DL-based reconstructions in CMR, focusing on the robustness, interpretability, clinical deployment, and potential for new methods.

## 1. Introduction

Over the last few decades, cardiac magnetic resonance (CMR) has emerged as an essential clinical tool to assess cardiovascular disease. CMR is non-invasive, free of ionizing radiation, and highly versatile, which enables a comprehensive assessment of the cardiovascular structure, function, flow, perfusion, viability, and tissue characterization [1,2]. For example, CMR is the gold standard for measuring the left ventricle (LV) ejection fraction, a cornerstone parameter for the diagnosis of heart failure and the timing of surgical interventions [3]. Additionally, late gadolinium enhancement (LGE) and T1/T2 mapping are widely used for cardiac viability and the detection of pathologies, such as diffuse myocardial fibrosis, fatty infiltration, or infarcted tissue [4,5,6,7,8,9,10,11]. Similarly, CMR flow imaging has revolutionized the care of patients with congenital heart disease by providing blood velocity and flow measurements in multiple planes, which allows for the detection and evaluation of intracardiac shunts and right- and left-lung flow splits [12,13,14].

However, one of the main drawbacks of CMR is the prolonged scan time. Magnetic resonance imaging (MRI) uses an inherently slow acquisition process, a disadvantage that compromises patient comfort and image quality, while limiting the achievable spatiotemporal resolution. CMR is especially challenging because clinicians require a high spatial and temporal resolution for accurate diagnoses. Additionally, CMR acquisitions must account for the inherent cardiac and respiratory motion to avoid motion artifacts [15,16,17,18].

To date, the most successful methods accelerate CMR data acquisition by reducing the amount of data acquired. Parallel imaging methods [19,20,21] exploit the redundancy of the data acquired through multi-channel coil arrays. Similarly, compressed sensing techniques [22] leverage the inherent compressibility (or sparsity) of MRI images. Both methods introduce aliasing artifacts that must be suppressed via a reconstruction algorithm to recover accurate images. For CMR, the combination of parallel imaging and compressed sensing (PICS) has achieved about 2–3x undersampling for 2D applications and 2–4x for 3D applications by exploiting both the multi-channel acquisition and the redundancy of spatiotemporal data, without compromising the accuracy or precision in the estimation of the cardiovascular parameters of clinical interest [23,24,25,26].

More recently, machine learning has disrupted the image reconstruction field and is rapidly becoming the state of the art. Deep learning (DL)-based reconstruction algorithms have outperformed conventional algorithms by yielding more accurate image reconstructions and enabling higher undersampling factors [27,28,29,30]. In compressed sensing, reconstruction algorithms remove aliasing artifacts by enforcing a prior image model based on sparsity [22,31,32]. DL-based methods are able to outperform these methods by learning the prior image model directly from historical data [33,34,35,36].

For CMR, DL-based reconstruction methods have rapidly surpassed PICS with increased undersampling rates and substantially reduced image reconstruction times. This advance has the potential to be translated clinically into improved image quality [37], for increases in the spatial and temporal resolution, and to significantly reduce the breath-hold durations, which is especially relevant for pediatric patients and adult patients with respiratory impairment [38,39]. The adaptation of DL-based reconstruction methods to CMR has brought significant technical developments. For example, multiple works have aimed to develop novel network architectures that are able to efficiently learn spatiotemporal priors [37,40,41,42]. Additionally, some works have been developed to tackle the specific challenges of CMR data, such as increased computational complexity [43,44] and a shortage of fully sampled ground-truth data for training [45,46].

In this work, we review the most current and central developments in DL-based reconstruction for CMR applications. In the first section, we overview the relevant image reconstruction theory. Then, we formally present several DL-based reconstruction methods. We first focus on the technical developments to adapt DL-based reconstruction to CMR data and then on the works developed for specific applications. Finally, we discuss the pitfalls, the future outlook of the field, and the present conclusions.

## 2. Image Reconstruction Theory

DL-based methods outperform conventional PICS image reconstruction by learning the prior image model directly from historical data. However, conventional image reconstruction is still relevant because DL-based reconstruction methods are heavily based on traditional PICS algorithms, as shown in Figure 1. We review the relevant image reconstruction theory, including the MR image models and image reconstruction algorithms, in order to contextualize the DL-based reconstruction methods in Section 3.

### 2.1. General Model

Compressed sensing [31,32] enables accelerated data acquisition by reducing the number of acquired *k*-space data required to obtain high-quality images via a pseudo-randomly undersampling scheme. This random undersampling strategy introduces incoherent aliasing artifacts into the images, which must be removed via a non-linear reconstruction algorithm to retrieve the high-quality image.

Image reconstruction can be formulated as an inverse problem [22] that aims to retrieve the image $x\in C^{NT}$ which is consistent with the acquired *k*-space data $y\in C^{M}$ and best matches a prior image model (e.g., sparsity or low rank). In CMR, *x* is a time series of images, with *N* voxels per time frame and *T* temporal frames. Additionally, *M* is the number of *k*-space locations acquired. Equation (Equation 1) formulates image reconstruction as an optimization problem.
(1)$$x^{*}=\arg\min_{x}\frac{1}{2}{∥Ax-y∥}_{2}^{2}+\lambda R\left(x\right),$$
where $A\in C^{M\times NT}$ is the MRI forward operator that includes the Fourier transform and the coil sensitivity maps from the multi-channel coil array, $R:C^{NT}\to R$ is a regularization function that enforces the prior, and λ is a tunable hyperparameter. The first least-squares term is denoted as a data consistency term because it enforces consistency with the acquired data, while the second term is the regularization term that enforces the prior image model.

For CMR, multiple choices of R that exploit the spatiotemporal redundancy have been proposed. Some models exploit sparsity in a sparsifying domain, such as temporal Fourier transform [24,50,51], wavelets [52], and total variation [53,54,55]. Likewise, other methods exploit low-rank structures [47,56,57] or a combination of both sparsity and low-rank models [58,59,60,61].

There are multiple algorithm options to solve the reconstruction problem described in Equation (Equation 1). If the proximal operator of R has a closed-form solution, e.g., ℓ1-wavelets [22], then an efficient option is to use a proximal gradient-type method [62] (e.g., a fast iterative shrinkage-thresholding algorithm [63]). The simplest proximal gradient algorithm alternates between a gradient step of the data consistency term and a proximal step on the regularization, as described in Equation ([Disp-formula FD2a-bioengineering-10-00334]) and Figure 1A.
(2a)$$\begin{array}{cc} z^{k+1} & =x^{k}-\alpha A^{H}\left(Ax^{k}-y\right), \end{array}$$
(2b)$$\begin{array}{cc} x^{k+1} & ={\mathrm{prox}}_{\alpha \lambda R}\left(z^{k+1}\right), \end{array}$$
where $x^{k}$ is the estimate of *x* at iteration *k*, $z^{k}$ is an auxiliary term at iteration *k*, α is the step size, and ${\mathrm{prox}}_{R}$ is the proximal operator [62] of the regularization function R.

Conversely, if the proximal operator does not have a closed-form solution, then more efficient methods would be the primal dual hybrid gradient [64] or the alternating direction methods of multipliers [65]. We describe the method denoted as half quadratic splitting [66], which is relevant for DL-based reconstruction, in Equation ([Disp-formula FD3a-bioengineering-10-00334]).
(3a)$$\begin{array}{cc} x^{k+1} & =\arg\min_{x}\frac{1}{2}{∥Ax-y∥}_{2}^{2}+\frac{\rho }{2}{∥Tx-z^{k}∥}_{2}^{2}, \end{array}$$
(3b)$$\begin{array}{cc} z^{k+1} & =\arg\min_{z}\lambda Q\left(z\right)+\frac{\rho }{2}{∥z-Tx^{k+1}∥}_{2}^{2}, \end{array}$$
where we consider a regularization function R that can be expressed as R(x)=Q(Tx), where Q is a convex function (usually a norm function) and *T* is a linear operator that represents the sparsifying transform (e.g., total variation).

### 2.2. Low Rank plus Sparse Model

A special and relevant formulation is the low rank plus sparse (L+S) decomposition model [26]. This model decomposes image *x* into two components: one low rank $x_{L}\in C^{NT}$, and one sparse $x_{S}\in C^{NT}$. Equation (Equation 4) presents the L+S reconstruction problem.
(4)$$x^{*}=\arg\min_{x}\frac{1}{2}∥A\left(x_{L}+x_{S}\right){-y∥}_{2}^{2}+\lambda_{L}∥x_{L}{∥}_{*}+\lambda_{S}{∥Tx_{S}∥}_{1},\;\;s.t.\;x=x_{L}+x_{S},$$
where $∥x_{L}{∥}_{*}$ is the nuclear norm of $x_{L}$, $∥Tx_{S}{∥}_{1}$ is the ℓ1-norm of $x_{S}$ in the *T* sparsifying transform domain. With this formulation, the aim is that $x_{L}$ captures the temporally correlated background, whereas $x_{S}$ captures the dynamic information on top of the background.

Equation (Equation 4) can be solved using an extension of the proximal gradient (Equation ([Disp-formula FD2a-bioengineering-10-00334])) as described in Equation ([Disp-formula FD5a-bioengineering-10-00334]).
(5a)$$\begin{array}{cc} x^{k+1} & =x^{k}-\alpha A^{H}\left(A\left(x_{L}^{k}+x_{S}^{k}\right)-y\right), \end{array}$$
(5b)$$\begin{array}{cc} x_{L}^{k+1} & ={\mathrm{prox}}_{\alpha \lambda_{L}{∥\cdot ∥}_{*}}\left(x^{k+1}-x_{S}^{k}\right), \end{array}$$
(5c)$$\begin{array}{cc} x_{S}^{k+1} & =T^{H}{\mathrm{prox}}_{\alpha \lambda_{S}{∥\cdot ∥}_{1}}\left(T\left(x^{k+1}-x_{L}^{k}\right)\right), \end{array}$$
where ${\mathrm{prox}}_{\alpha \lambda_{L}{∥\cdot ∥}_{*}}$ is the proximal operator of the nuclear norm, i.e., the singular value soft-thresholding operation, and ${\mathrm{prox}}_{\alpha \lambda_{S}{∥\cdot ∥}_{1}}$ is the proximal operator of the ℓ1-norm, i.e., the soft-thresholding operation.

### 2.3. Partial Separability Model

Lastly, another formulation of interest uses the partial separability model [67]. This model assumes that the temporal images can be represented as linear combinations of a smaller number of temporal components. Therefore, the time series of images $x\in C^{NT}$ can be stacked to form a Caseroti matrix $X\in C^{N\times T}$, which will be rank deficient with rank R<T≪N. Then, *X* can be expressed as a multiplication of two matrices $U\in C^{N\times R},V\in C^{T\times R}$, where *U* contains the image-domain basis and *V* the temporal basis as shown in Equation (Equation 6):(6)$$X=\left[\begin{array}{ccc} x_{1,1} & \cdots & x_{1,T}\\ \vdots & \ddots & \vdots \\ x_{N,1} & \cdots & x_{N,T} \end{array}\right]=UV^{H}=\left[\begin{array}{ccc} U_{1,1} & \cdots & U_{1,R}\\ \vdots & \ddots & \vdots \\ U_{N,1} & \cdots & U_{N,R} \end{array}\right]\left[\begin{array}{ccc} V_{1,1} & \cdots & V_{1,T}\\ \vdots & \ddots & \vdots \\ V_{R,1} & \cdots & V_{R,T}. \end{array}\right]$$

The reconstruction problem becomes
(7)$$X^{*}=\arg\min_{X}\frac{1}{2}{∥A\left(UV^{H}\right)-y∥}_{2}^{2}+\lambda R\left(U,V\right),\;\;s.t.\;X=UV^{H},$$
where $A:C^{N\times T}\to C^{M}$ is the MRI linear measurement operator, and R is the regularization function for *U* and *V*. There are multiple options for R that include sparse [60] or low-rank [47,48] regularizations.

Regarding the solver for Equation (Equation 7), some works pre-compute the temporal basis *V* using a calibration procedure and solve only for *U* [56,60]. A relevant case is when we simultaneously estimate *U* and *V*, where we can use the alternating minimization procedure described in Equation ([Disp-formula FD8a-bioengineering-10-00334]) [47,48] and Figure 1B.
(8a)$$\begin{array}{cc} U^{\left(k+1\right)} & =\arg\min_{U}\frac{1}{2}{∥A\left(UV^{(k)H}\right)-y∥}_{2}^{2}+\lambda R\left(U,V^{\left(k\right)}\right), \end{array}$$
(8b)$$\begin{array}{cc} V^{\left(k+1\right)} & =\arg\min_{V}\frac{1}{2}{∥A\left(U^{\left(k\right)}V^{H}\right)-y∥}_{2}^{2}+\lambda R\left(U^{\left(k\right)},V\right). \end{array}$$One important advantage of the partial separability model is the computational efficiency. Because the rank *R* is usually significantly lower than the dimensions of *X*, performing a reconstruction on the basis of *U* and *V* is more computationally efficient, especially when the regularization function R does not require the explicit formation of *X*, as demonstrated in Ong et al. [48].

## 3. Deep Learning-Based Reconstruction

With the rapid adoption of machine learning in multiple areas of science, DL rapidly became a powerful tool for MR image reconstruction. While conventional compressed sensing assumes a generic sparse image prior, DL-based reconstruction methods enforce a data-driven image prior tailored for each application by learning the prior directly from historical data. Due to their success in tasks such as image segmentation and classification, the first DL-based reconstruction methods applied common architectures, such as U-Nets and residual networks [68,69,70,71]. However, soon more specialized architectures were developed.

In this section, we describe the general learning and evaluation procedures. Additionally, we review the main network architectures developed for CMR reconstruction, with the main focus on applications to cardiac cine MRI. Cardiac cine is the most widely used CMR technique in the clinic and, thus, the most commonly tackled CMR reconstruction problem [1]. We leverage the extensive bibliography of works developed for this particular application to elaborate on the most relevant network architectures. Furthermore, we emphasize a specialized type of framework denoted as unrolled networks, which are heavily based on conventional reconstructions.

### 3.1. Learning and Evaluation Procedure

The most common approach to learn a reconstruction network is through supervised learning. In this approach, it is ideal to have a set of fully-sampled datasets that will be used as ground truth for training the network. These datasets are split into three subsets: (1) training; (2) validation; and (3) testing, e.g., a standard split would be 70/20/10% [28]. The training subset is used to train the network (i.e., to calculate the gradients and update the network weights). The validation subset is used for hyperparameter tuning and to assess generalization. The test subset is used after training to properly assess the performance of the network and report image quality metrics relative to the fully sampled ground-truth datasets.

The training procedure aims to minimize a loss function L similar to Equation (Equation 9).
(9)$$L\left(\Phi \right)=\sum_{i}\left|\right|G_{\Phi }\left(x_{i}\right)-m_{i}{\left|\right|}_{2}^{2},$$
where $m_{i}$ is the *i*-th sample of the fully sampled training subset, $x_{i}$ is the corresponding retrospectively undersampled image ($x_{i}=A^{H}M_{i}Am_{i}$, where $M_{i}$ is a *k*-space undersampling mask), and G is the reconstruction network parameterized by the weights Φ. In this example, we used an $\ell_{2}$-norm loss function, but other options include $\ell_{1}$-norm, SSIM (structural similarity index), and perceptual loss functions [28,72]. The loss function can be minimized using a stochastic gradient descent algorithm, such as Adam [73].

The success of the training procedure and, thus, the learned reconstruction network depends highly on the quality, authenticity, and diversity of the training subset [36]. First, although *k*-space data are typically not readily available, advanced reconstruction networks need access to the raw *k*-space measurements, which provide information about the phase of the image and the coil sensitivity maps. Neglecting or simulating (e.g., from DICOM magnitude-only images) this information has led to artificial over-optimistic results [74]. Furthermore, data pre-processing steps such as normalization can help to greatly improve the training convergence. Data can be normalized using the $\ell_{2}$-norm of the central *k*-space measurements, such that all data have approximately the same scale [30,36]. Finally, a common approach to improve diversity is data augmentation. Different undersampling masks $M_{i}$, flipping, and transposing the images are some common transformations used for data augmentation [75].

The validation subset is used for the generalization assessment and hyperparameter tuning. During training, a validation loss can be estimated using Equation (Equation 9) on the validation subset at regular intervals. The aim is to assess how well the network has generalized and performed with unseen data. Furthermore, multiple training runs with different hyperparameter values can be performed. The performance on the validation subset between these runs can be compared to select the best hyperparameter configuration.

Finally, after training, the evaluation procedure assesses the network performance on the testing subset. In general, the performance is evaluated by comparing the reconstructed images with the fully sampled ground-truth images in the datasets through an image quality metric, such as the NRMSE (normalized root mean squared error), pSNR (peak signal-to-noise ratio), and SSIM [30].

### 3.2. Unrolled Networks

Currently, the state-of-the-art methods use the unrolled network [33,34,35] framework, which is heavily based on conventional image reconstruction. These networks resemble the traditional iterative optimization algorithms by alternating between data consistency and regularization (Figure 1). The difference lies in the regularization step, where the algorithms enforce data-driven priors via a deep neural network. Next, we review some relevant CMR works that use this framework.

#### 3.2.1. Neural Proximal Gradient Descent

The neural proximal gradient descent [35] or variational network [33] resembles the proximal gradient algorithm in Equation ([Disp-formula FD2a-bioengineering-10-00334]) and is described in Equation ([Disp-formula FD10a-bioengineering-10-00334]) and Figure 1A.
(10a)$$\begin{array}{cc} x^{t+1} & =x^{t}-\alpha A^{T}\left(Az^{t}-y\right), \end{array}$$
(10b)$$\begin{array}{cc} z^{t+1} & =D_{\Phi }\left(x^{t+1}\right), \end{array}$$
where $D_{\Phi }$ is a regularization function performed by a deep neural network parameterized by Φ. This architecture resembles the proximal gradient descent (Equation ([Disp-formula FD2a-bioengineering-10-00334])) by alternating gradient steps of the data consistency term and regularization steps. In this case, instead of a proximal step, the regularization is performed by a neural network $D_{\Phi }$ that enforces a data-driven prior learned directly from historical data through the learning procedure (Section 3.1).

For CMR, Sandino et al. [37] proposed DL-ESPIRiT, an unrolled network with the neural proximal gradient architecture and an extended coil sensitivity map operator to address field-of-view (FOV) limitations. In the data consistency step, DL-ESPIRiT used an augmented forward model with multiple sets of sensitivity maps, based on ESPIRiT, to flexibly represent overlapping tissue in reduced FOV acquisitions, which are common in cine imaging. Additionally, in the regularization step, the algorithm used a novel CNN architecture with separable 2D spatial + 1D temporal convolutions to learn spatiotemporal priors more efficiently. DL-ESPIRiT showed improved agreement compared to standard PICS and conventional parallel imaging, both qualitatively (sharpness) and quantitatively through cardiac function indices (stroke volume and ejection fraction). Furthermore, in the follow-up clinical study [38], DL-ESPIRiT yielded an approximate three-fold reduced scan time (0.9±0.3 min versus 3.0±1.9 min) with a slightly lower image quality (only 0.5 less on a five-point Likert scale) than the clinical standard.

Similarly, Kustner et al. [40] used the neural proximal gradient architecture for their CINENet, a network focused on 3D cardiac cine imaging. In the regularization step, CINENet used 4D U-Nets with cascaded separable 3D spatial + 1D temporal convolutions. This novel architecture enabled the acquisition of 3D cine imaging in a single breath-hold (less than 10 s). For comparison, they performed a standard 2D multi-slice acquisition, which required multiple breath-holds and a total acquisition time of 260 s. Furthermore, the authors showed an improved image quality and contrast compared to PICS and good agreement in the LV functional parameters (end-systolic volume, end-diastolic volume, and ejection fraction) with the clinical gold-standard 2D cine imaging.

#### 3.2.2. Model-Based Reconstruction Using Deep-Learned Priors

Another relevant method is the MOdel-based reconstruction using Deep-Learned priors (MoDL) [34], which resembles the half quadratic splitting (Equation ([Disp-formula FD3a-bioengineering-10-00334])), and is described in Equation ([Disp-formula FD11a-bioengineering-10-00334]):
(11a)$$\begin{array}{cc} x^{t+1} & =\arg\min_{x}\frac{1}{2}{∥Ax-y∥}_{2}^{2}+\frac{\rho }{2}{∥\Psi x-z^{t}∥}_{2}^{2}, \end{array}$$
(11b)$$\begin{array}{cc} z^{t+1} & =D_{\Phi }\left(x^{t+1}\right), \end{array}$$
where $D_{\Phi }$ is, likewise, a regularization function implemented as a deep neural network. The MoDL architecture is similar to the neural proximal gradient descent, with the principal difference being the data consistency step, where a quadratic optimization problem is solved instead of performing a simple gradient step.

For CMR, Biswas et al. [41] introduced a framework (MoDL-SToRM) [41] that combines the MoDL architecture with conventional image regularization penalties to reconstruct free-breathing and ungated cardiac MRI data from highly undersampled (R = 50x) multi-channel measurements. Their method incorporates the MoDL-based [34] reconstruction shown in Equation ([Disp-formula FD11a-bioengineering-10-00334]) into the SmooThness Regularization on Manifolds (SToRM) [76] framework, which assumes that the images in the free-breathing acquisitions lie on a smooth and low-dimensional manifold parameterized by the cardiac and respiratory phase and navigator radial spokes to capture the structure of the manifold. Consequently, the proposed training loss function synergistically combines the data consistency term, convolutional neural network (CNN) regularization prior, and SToRM prior. In particular, the CNN prior captures the local and population-generalizable dependencies and the SToRM prior utilizes patient-specific information, including cardiac and respiratory patterns. On highly undersampled (R = 50x) simulated and prospectively acquired 2D+time datasets, the authors showed that the MoDL-SToRM framework provides the most accurate reconstructions compared to MoDL-only and SToRM-only baselines with the same number of frames. In terms of the scan time, MoDL-SToRM reconstructions from 8.2 s of MRI data per slice are comparable to SToRM reconstructions from 42 s of MRI data per slice, resulting in a 5x reduction in scan time for the same image quality. In addition, the inference time for the proposed method is roughly 30 s for 200 frames on a P100 GPU, which is significantly faster than CS reconstructions.

#### 3.2.3. Deep Low Rank plus Sparse

Huang et al. [42] adapted the low rank plus sparse (L+S) image model (Section 2.2) and incorporated a deep learning prior for the sparse component. The method is described in Equation ([Disp-formula FD12a-bioengineering-10-00334]):
(12a)$$\begin{array}{cc} x^{k+1} & =x^{k}-\alpha A^{H}\left(A\left(x_{L}^{k}+x_{S}^{k}\right)-y\right) \end{array}$$
(12b)$$\begin{array}{cc} x_{L}^{k+1} & ={\mathrm{prox}}_{\alpha \lambda_{L}{∥\cdot ∥}_{*}}\left(x^{k+1}-x_{S}^{k}\right), \end{array}$$
(12c)$$\begin{array}{cc} x_{S}^{k+1} & =D_{\Phi }\left(x^{k+1},x_{L}^{k}\right), \end{array}$$
where $D_{\Phi }\left(x^{k+1},x_{L}^{k}\right)$ is a regularization deep neural network. Similarly to the original method, the image is decomposed into two components: one low rank ($x_{L}$) and another sparse ($x_{S}$). The difference is in the regularization step of the sparse component where, instead of a proximal step, a deep convolutional neural network $D_{\Phi }$ applies a data-driven regularization.

L+S-Net achieved an improved image quality in terms of the NRMSE, pSNR, and SSIM when retrospectively evaluated and compared to the conventional L+S. Additionally, L+S-Net showed an improved image quality, i.e., reduced streaking artifacts and spatial blurring, when prospectively evaluated.

#### 3.2.4. Deep Subspace Learning

Sandino et al. [49] leveraged the computational efficiency of the partial separability model (Section 2.3 and Figure 1B) to propose DL-Subspace, an unrolled network focused on reducing memory usage and reconstruction time. In the partial separability model, instead of solving directly for image *x*, we aim to solve for the lower-dimensional subspace of image *x*, i.e., the spatial basis *U* and temporal basis *V*. Sandino et al. [49] incorporated deep learning priors to this model by using the MoDL framework. The simplified algorithm is presented in Equation ([Disp-formula FD13a-bioengineering-10-00334]).
(13a)$$\begin{array}{cc} U^{\left(k+0.5\right)} & =\underset{U}{\arg\min}{∥y-A\left(UV^{(k)H}\right)∥}_{F}^{2}+\lambda_{u}{∥U^{\left(k\right)}-U∥}_{F}^{2}, \end{array}$$
(13b)$$\begin{array}{cc} U^{\left(k+1\right)} & =D_{\Phi_{u}}\left(U^{\left(k+0.5\right)}\right) \end{array}$$
(13c)$$\begin{array}{cc} V^{\left(k+0.5\right)} & =\underset{V}{\arg\min}{∥y-A\left(U^{\left(k+1\right)}V^{H}\right)∥}_{F}^{2}+\lambda_{v}{∥V^{\left(k\right)}-V∥}_{F}^{2}, \end{array}$$
(13d)$$\begin{array}{cc} V^{\left(k+1\right)} & =D_{\Phi_{v}}\left(V^{\left(k+0.5\right)}\right), \end{array}$$
where $D_{\Phi_{u}}$ and $D_{\Phi_{v}}$ are 2D and 1D CNNs that denoise the spatial and temporal basis, respectively. Equation ([Disp-formula FD8a-bioengineering-10-00334]) has been divided into a MoDL data consistency step (Equation ([Disp-formula FD13a-bioengineering-10-00334])) and a data-driven regularization step (Equation ([Disp-formula FD13b-bioengineering-10-00334])). Similarly, Equation ([Disp-formula FD8b-bioengineering-10-00334]) has been divided into data consistency (Equation ([Disp-formula FD13c-bioengineering-10-00334])) and regularization (Equation ([Disp-formula FD13d-bioengineering-10-00334])). The DL-Subspace network showed improved computational efficiency in terms of the memory usage and inference speed versus MoDL [34] and DL-ESPIRiT [77]. Additionally, DL-Subspace outperformed the other networks in terms of the SSIM and PSNR with respect to the fully sampled reference.

### 3.3. Other Networks

Exploring the spatiotemporal correlations of dynamic signals proves crucial for enhanced acceleration in cardiac imaging. Manifold learning is a useful technique for leveraging such correlations, which assumes dynamic signals in high-dimensional space are neighboring points on smooth, low-dimensional manifolds. Manifold-Net [78] is a method based on manifold learning that introduces a deep learning model to capture the temporal redundancy in reconstructing dynamic cardiac data from undersampled measurements. Specifically, it introduces a fixed, low-rank tensor manifold for temporal correlations, then models the dynamic reconstruction as a general CS-based optimization problem on the manifold. This is solved via Riemannian optimization, and the iterations of the optimization are unrolled into a neural network. This unrolled implementation is useful for reducing the computation time and eliminating the need for ad hoc parameter selection. Manifold-Net is demonstrated to yield improved quantitative measurements in terms of the mean squared error (MSE), PSNR, and SSIM for up to a 12-fold acceleration of cardiac data with retrospective electrocardiogram-gated segmented balanced steady-state free precession (bSSFP) with breath-holding. The authors also report that Manifold-Net preserves its high reconstruction accuracy over different sampling masks based on pseudo-radial, pseudo-spiral, and VISTA [79], and over limited training data.

Another approach that allows spatial resolution improvements without sacrificing scan times is super resolution. Super resolution (SR) refers to the recovery of high-resolution images from the low-resolution counterparts. Traditional image upscaling approaches include bicubic interpolation methods and zero padding in the Fourier domain which achieves sinc interpolation [80,81]. Recent advances in DL have also shown improvements in SR techniques. In particular, convolutional neural network-based SR has demonstrated a superior quality over traditional methods in computer vision-related fields [82,83,84,85]. SR has the potential to reduce the scan time and/or increase the temporal resolution for CMR.

Evan et al. proposed a DL SR method [86] to enhance the spatial detail of cardiac MRI from low-resolution acquisitions that are faster to acquire. For training, the authors used retrospectively collected short-axis bSSFP cine images from 400 examinations, from both 1.5T and 3T scanners. These data were treated as the ground truth and then only a portion of the central *k*-space was retained to simulate low-resolution acquisitions. They used a shallow (SRNet) and a deeper (modified U-Net [87]) neural network in single-frame (k) and multi-frame (kt) SR settings with an L1 + SSIM hybrid loss function. The single-frame networks had 2D convolutions, whereas the multi-frame networks consisted of 3D convolutions with the temporal domain in the third dimension. Retrospective evaluations across a wide range of upsampling factors demonstrated that all of their CNN methods quantitatively and qualitatively outperformed the conventional zero-padding and bicubic interpolation strategies. The authors also tested their SR approach on prospectively acquired scans and reported the comparable LV end-diastolic volume, LV end-systolic volume, LV stroke volume, and LV ejection fraction values to those from full-resolution images while visually improving anatomic details.

Baty and Grau presented another SR approach [88] that learns the mapping between low-resolution dynamic short-axis images and high-resolution long-axis views. They used a modified U-Net [87] architecture with convolutional long short-term memory layers at the first and second levels to take the dynamic aspect of the cine sequence into account via the high spatial frequency features across multiple time steps. Their results demonstrated that their proposed network yields improved quantitative and qualitative results compared to the standard static U-Net baseline and the traditional interpolation approach, highlighting the importance of temporal context for the super resolution of cardiac cine MRI sequences.

An alternative approach for DL SR has been proposed by Zhao et al., denoted as Synthetic Multi-Orientation Resolution Enhancement (SMORE) [89]. SMORE is a self-supervised model for MRI acquisitions with thick slices. This corresponds to images acquired with a high in-plane resolution but a poor through-plane resolution, enabling a reasonable acquisition time in general. The SMORE model is based on the idea that high-resolution in-plane slices can be used to compile paired low-resolution/high-resolution training data by down-sampling in a self-supervised fashion. For this purpose, SMORE applies a filter h(x) in the *k*-space containing a rect filter together with an anti-ringing filter on in-plane slices to yield a through-plane low resolution. Generated low-resolution/high-resolution data are then used to train the self super-resolution network. The authors demonstrate SMORE on a cardiac MRI to perform a super resolution from a 1.1 × 1.1 × 5 mm${}^{3}$ to an isotropic 1.1 mm${}^{3}$ resolution and to explore the visualization of myocardial scarring from cardiac LV remodeling after myocardial infarction. The SMORE method yields enhanced scar visualization with improved image clarity.

## 4. CMR-Specific Challenges

Due to the particular characteristics of CMR, DL-based reconstructions present some specific challenges. For example, CMR images have higher dimensionality (spatial plus temporal dimensions), which increases the computational cost of the reconstruction. Additionally, due to the same increased dimensionality, some CMR techniques have scarce or unavailable access to fully sampled ground-truth datasets for training. In this section, we review some works specifically developed to tackle these problems.

### 4.1. Increased Dimensionality

Deep unrolling algorithms for reconstruction involve multiple iterations of a neural network that is typically trained end-to-end using backpropagation. However, this approach requires large memory and computation power in calculating gradients and storing intermediate activations. The requirements also increase linearly with the measurement count and the number of network layers. High-dimensional reconstruction tasks can be particularly challenging, with exceptionally large memory requirements resulting in a reduction in the number of unrolled iterations at the expense of lower accuracy. Memory-efficient training frameworks aim to address the computational challenges of deep unrolling algorithms by exploring the trade-offs between memory usage and performance. These methods can take various forms such as constrained network families that enable reduced memory usage or modified training strategies that limit the amount of memory occupied by the network at any given time.

An important approach to address the limitations of large-scale DL reconstruction is Memory-Efficient Learning (MEL) [44]. MEL treats the full graph of the unrolled implementation as a series of smaller sequential graphs, which allows the network to be forward propagated to obtain the final output without computing the network gradients. MEL then uses invertible CNN models across the unrolls to recompute each smaller auto-differentiation graph from the network’s output in the reverse order. This approach allows MEL to only store a single layer in memory at a time, reducing the required memory by a factor of N. Notably, the computation required to invert each layer only marginally increases the backpropagation run time. MEL has been trained and evaluated on retrospectively and prospectively undersampled 3D knee and 2D+time cardiac cine MRI. It has been shown to improve quantitative metrics and reconstruct finer details with sharper edges and more continuous textures with a higher overall image quality for both 2D+time and 3D+time cardiac cine MRI. These performance improvements were achieved using a single 12 GB GPU.

Greedy LEarning for Accelerated MRI (GLEAM) [43] is another memory-efficient learning algorithm that splits the reconstruction network into independent modules and performs gradient-based optimization for each decoupled module in a greedy manner. Each module computes its own forward pass, updates its gradients based on a local loss, and passes the output to the next module. The intermediate activations are then discarded from the GPU memory. This allows GLEAM to only store a single module in memory at a time, reducing the GPU memory requirements to the amount required by a single module. Additionally, GLEAM can be extended to a distributed setting, enabling gradient updates for each module to be performed on different GPUs in parallel to speed up training. Unlike invertible networks [44,90,91], GLEAM does not suffer from numerical instabilities when there are a large number of layers because it decouples each module and relies on backpropagation instead of inverting individual modules. GLEAM has been evaluated on multi-coil knee, brain, and dynamic cardiac cine MRI datasets with multiple reconstruction network architectures. It has been shown to improve both qualitative and quantitative metrics over end-to-end learning baselines while also reducing the memory footprint and speeding up training compared to existing memory-efficient learning methods. On a 2D+time cardiac cine dataset, GLEAM was able to achieve a 1.8 dB pSNR improvement compared to traditional end-to-end learning with backpropagation by allowing three times as many unrolled iterations.

### 4.2. Limited Training Data

As described in the previous sections, supervised learning approaches have been successfully employed for the fast and high-quality reconstruction of undersampled MRI. One drawback of these methods is the reliance on fully sampled ground-truth data in the training datasets. This dependence poses a bigger problem for dynamic MRI as fully sampled data are typically unavailable in settings like cardiac imaging due to scan time constraints. Therefore, machine learning techniques that remove or reduce the requirement for fully sampled data are crucial for the success of DL-based reconstruction in dynamic MRI applications.

Deep Image Prior (DIP) [45] is an approach that solves inverse problems such as image denoising, inpainting, and SR in an unsupervised fashion (i.e., without the need of labeled training data). DIP optimizes the parameters of an untrained neural network that maps a fixed latent vector to the measurements. Note that DIP does not need a pre-trained network or an image database [45] and relies on the implicit bias of the CNN architecture. Yoo et al. proposed a Time-Dependent Deep Image Prior (TD-DIP) [92] approach that adapts the DIP method to dynamic MRI by optimizing an untrained neural network that maps the sequence of latent vectors to radial spoke-shared measurements. In addition, the authors constrain the latent vectors to a one-dimensional manifold so that the CNN learns to encode the temporal variations in the cine images into the spatial closeness of the latent samples. In this way, the temporal dependencies of the dynamic measurements, such as the nearly periodic cardiac motion, can be encoded more effectively. In order to provide more flexibility, the authors also include a Mapping Network (MapNet) that maps the fixed manifold into a more expressive latent space. They demonstrated that the TD-DIP method qualitatively and quantitatively outperforms the compressed sensing baselines with experiments on retrospective and prospective cardiac datasets. Because the TD-DIP approach captures the temporal dynamics with the manifold design, it also enables continuous dynamic reconstruction in the sense that intra-frame images can be recovered by traversing between consecutive latent vectors in the manifold.

Based on the success of recent self-supervised techniques for static MRI reconstruction without fully sampled ground-truth data, Acar et al. introduced a self-supervised learning framework for training deep architectures for dynamic cardiac MRI reconstruction [46]. The framework is depicted in Figure 2. Given that the model has no access to a fully sampled *k*-space, this framework splits the undersampled *k*-space data into two non-overlapping segments and uses the first segment to enforce data consistency and the second segment to learn the network weights. Their systematic assessment of the amenability of models of varying complexity on an open-access multi-coil *k*-space dataset for cardiovascular MRI demonstrated that a compact unrolled CNN architecture with bidirectional recurrent connections exhibits enhanced robustness.

## 5. Application-Specific Methods

### 5.1. Cardiovascular Blood Velocity and Flow Quantification

Phase contrast MRI (PC-MRI) provides quantitative measurements of the blood velocity and flow throughout the cardiovascular system. Two-dimensional PC-MRI (2D-Flow) is widely considered to be the clinical gold standard [93,94] for the non-invasive and time-resolved quantification of the blood velocity (cm/s) and flow (mL/beat) at a single slice location, whereas 3D+time PC-MRI (4D-Flow) is an increasingly impactful clinical tool used to visualize the blood velocity and flow patterns in a time-resolved 3D volume encompassing the entire chest. Both 2D-Flow and 4D-Flow provide clinicians with important hemodynamic parameters that aid in the diagnosis of many cardiovascular diseases, including valvular stenosis and pressure gradients in addition to regurgitation, right/left-lung flow splits, and shunts (flow). However, the spatiotemporal requirements for the accurate and precise measurements of the blood velocity and flow place an upper bound on the scan time [95,96].

DL-based reconstruction methods for both 2D- and 4D-Flow have recently emerged as a promising alternative, which aim to drastically reduce the scan time without reconstruction times that challenge the clinical workflow, which is typical of most CS techniques. Table 1 provides an overview of the currently published literature for DL-based reconstruction methods for 2D- and 4D-Flow.

Vishnevskiy et al. were the first to describe a DL-based reconstruction for the undersampled 4D-Flow [97]. They trained a deep variational neural network [33] using healthy subjects (n = 11). They retrospectively undersampled one patient dataset and prospectively undersampled seven volunteer datasets using a Cartesian pseudo-radial golden-angle sampling pattern with 12.4 ≤ R ≤ 13.8 [104]. Compared to R = 2 PI, their prospectively deployed DL-based reconstruction provided comparable measurements of peak velocity (bias = 1.6%, limits of agreement (LOA) = [−11.2, 8.1]%), and peak flow (0.05%, [−9.8, 9.7]%) in the ascending aorta (aAo) while also demonstrating a reduced bias and tighter limits of agreement (LOA) compared to the CS [105]. A representative patient dataset is included in Figure 3, highlighting the improvement in their network over the CS.

Haji-Valizadeh et al. were the first to describe a DL-based reconstruction for the real-time 2D-Flow [98]. They trained a 3D U-Net using synthetic real-time 2D-Flow datasets, generated from 510 ECG-gated breath-hold patient datasets, to separately filter aliasing artifacts from radial real-time velocity-compensated and complex-difference images. They prospectively undersampled 21 patient datasets and compared the ECG-gated 2D-Flow and the proposed real-time 2D-Flow sequence with radial *k*-space sampling for R = 28.8. Compared to the ECG-gated 2D-Flow, their prospectively deployed real-time radial 2D-Flow sequence with a DL-based reconstruction provided comparable measurements of the cardiac output (−0.8 [−1.7, 0.1] L/min), stroke volume (−9.5 [−24.2, 5.3] mL), and peak mean velocity (−4.4 [−19.2, 10.4] cm/s) in the aAo while also demonstrating a reduced bias and tighter LOA compared to the CS [106].

Cole et al. were the first to investigate the impact of complex-valued CNNs for the reconstruction of an undersampled 2D-Flow [99]. They trained two U-Net networks using end-to-end complex-valued CNNs as well as real-valued CNNs, using 180 fully sampled 2D-Flow pediatric patient datasets. They retrospectively undersampled 44 patient datasets, using variable density pseudo-random Poisson-disc undersampling for R ≤ 6. Compared to the real-valued CNN, the complex-valued CNN provided a higher PSNR (33.5 ± 5.9 vs. 32.3 ± 5.5), lower NRMSE (0.21 ± 0.1 vs. 0.22 ± 0.08), and higher SSIM (0.88 ± 0.08 vs. 0.86 ± 0.08) in addition to reducing errors in the measurement of the flow (6.3% vs. 7.2%) and peak velocity (5.3% vs. 28.2%).

Jaubert et al. proposed a DL-based reconstruction using complex-valued images for a spiral real-time 2D-Flow [100]. They trained a 3D U-Net using synthetic real-time spiral 2D-Flow datasets, generated from 520 ECG-gated breath-hold patient datasets. They prospectively undersampled 20 patient datasets and compared the ECG-gated 2D-Flow and the proposed real-time 2D-Flow sequence with uniform density spiral undersampling for R = 18. Compared to the ECG-gated 2D-Flow with R = 2 PI, their prospectively deployed real-time radial 2D-Flow sequence with DL-based reconstruction provided comparable measurements of the stroke volume (−0.8 [−10.5, 8.9] mL) and peak mean velocity (−2.5 [−13.7, 8.7] cm/s) in the aAo while also demonstrating a reduced bias and tighter LOA for the stroke volume but not the peak mean velocity, compared to the CS [107]. They later expanded their original work to demonstrate real-time 2D-Flow measurements, including visualization on the scanner of the beat-to-beat heart rate, stroke volume, and cardiac output with a mean latency of 622 ms [101]. Importantly, no significant differences were observed between the reference R = 2 PI methods at rest (−0.21 ± 0.50 L/min, *p* = 0.25) or the CS at peak exercise (0.12 ± 0.48 L/min, *p* = 0.46). A representative patient dataset is included in Figure 4, highlighting the improvement in their U-Net over CS.

Oscanoa et al. demonstrated a DL-based reconstruction for a highly undersampled 2D-Flow, which made use of a complex difference estimation to promote image sparsity [39]. They trained an unrolled network using 155 fully sampled 2D-Flow pediatric patient datasets. They prospectively undersampled five patient datasets and compared R = 2 PI with variable density pseudo-random Poisson-disc undersampling for R ≤ 8. Compared to R = 2 PI, their prospectively deployed sequence with DL-based reconstruction provided comparable measurements of the peak velocity (−3.6 [−5.6, 4.5]%) and total flow (6.1 [−11.6, 19.4]%) in the aAo and pulmonary artery (PA) while also demonstrating a reduced bias and tighter LOA for acceleration rates up to 8x, compared to the CS [108].

Kim et al. demonstrated a DL-based reconstruction for the 4D-Flow that produced three directional velocities from only three acquired flow encodings, without requiring the conventional fourth reference scan to measure the background phase [102]. They used the complex images from four velocity encodings (flow-compensated reference plus x, y, and z velocities) of 144 neurovascular patient datasets to train a 3D U-Net to predict the x, y, and z velocities. They took 14 3D radial undersampled 4D-Flow patient datasets and compared a conventional 4-point to the proposed DL-based 3-point reconstruction with the native undersampling and also with further undersampling applied retrospectively using R ≤ 6. Without additional undersampling, the DL-based 3-point reconstruction provided comparable measurements of the total flow (R${}^{2}$ = 0.98, slope = 0.95, *p* < 0.001, 95%–CIs = [0.93, 0.97]) and peak velocity (R${}^{2}$ = 0.98, slope = 0.96, *p* < 0.001, 95%–CIs = [0.93, 0.99]) compared to the 4-point reconstruction, across several notable neurovascular arteries. With R = 6 additional undersampling, the 3-point DL-based recon underestimated the velocities, relative to the 4-point reconstruction, by about 3.1% of the prescribed velocity encoding strength.

Nath et al. demonstrated DL-based residual attention reconstruction for the 4D-Flow [103]. They trained 2D U-Nets separately for each 4D-Flow velocity encoding direction using 2D complex image slices extracted from the 4D-Flow volume at each temporal frame and slice position, using 18 fully sampled volunteer datasets. They retrospectively undersampled two fully sampled patient datasets using variable density pseudo-random Poisson-disc undersampling for R = 2.5, 3.3, and 5. Compared to the fully sampled datasets, their proposed DL-based reconstruction with R = 2.5 provided comparable measurements of the peak velocity (10.3 ± 5.1 cm/s) and total flow (18.2 ± 4.1 mL/s), with larger errors for R = 3.3 and R = 5, but did demonstrate a reduced bias and tighter LOA compared to the three state-of-the-art DL-based reconstructions and CS [109] across all undersampling factors.

### 5.2. Late Gadolinium Enhancement

Late gadolinium enhancement (LGE) imaging is a technique that measures contrast agent extravasation into the myocardium and is considered the gold standard for diagnosing myocardial scar tissue. Accelerating LGE acquisitions is particularly important, as there is a limited time window to capture an image before the contrast dynamics change and decrease the diagnostic quality of the images acquired. Recent DL reconstruction works have focused on providing whole heart 3D volumes with significant undersampling to provide full-heart high-resolution scar characterization in a clinically viable scan time.

El-Rewaidy et al. developed a U-Net-based method for reconstructing 3D LGE volumes from Cartesian undersampled datasets [110]. The network presented used fully complex convolutions rather than real-valued convolutions with real and imaginary channels. Additionally, a batch normalization layer was used that better preserved the magnitude and phase relationships in the complex data. The image quality was scored by expert reviewers, where the differences between the complex-valued network and a reference CS reconstruction were insignificant. In the same comparison, the real-valued network was found to score significantly worse. The scar extent was also compared between the reference CS reconstruction and the complex-valued network, where a very good correlation ($R^{2}>0.97$) and very small bias were seen between the methods. The computation time of the neural networks was over 300× shorter than the reference CS reconstruction. The work showed that DL reconstruction networks could be applied to significantly speed up undersampled reconstructions without loss in the image quality or diagnostic applicability (Figure 5).

Yaman et al. designed a DL reconstruction method using physics-based networks to reconstruct 3D LGE images with random Cartesian undersampling [111]. The method used an unrolled network that combined data consistency blocks and 3D convolutional blocks for regularization. The data were trained in a self-supervised manner, so no fully sampled data were needed. This was achieved by splitting the *k*-space data into two sets and using one set for the data consistency and the other for a loss function requiring the *k*-space data to match the left-out set. Based on clinical reviewer scoring, the method allowed for better image quality, with undersampling ratios up to R = 6 when compared to a standard CS reconstruction.

Ven der Velde et al. investigated the use of a DL reconstruction on LGE image quality, as well as the performance of automatic scar segmentation methods [112]. A CNN-based network architecture was used with a tunable denoising parameter, from 0 to 100%. The method was tested on 2D LGE images with different noise reduction factors, where it was seen that the DL recon significantly improved the image quality based on expert scoring. The scar size was also quantified, where it was observed that standard deviation (SD)-based automatic segmentation methods overestimated the scar size when compared to manual segmentation. This was attributed to the new noise characteristics from the recon changing the thresholding levels when based off the SD. Different segmentation algorithms, such as a full width at half maximum method, did not show these same biases with the DL recon data.

### 5.3. Tissue Characterization

Cardiac tissue characterization generally involves dynamic imaging in combination with RF events to characterize tissue by T1 and T2. These techniques have shown to be clinically important biomarkers for a range of cardiac pathologies. Acceleration is important to obtain more temporal information needed to fit T1 and T2 and to obtain whole heart coverage in a reasonable amount of time. Because these techniques can span three independent temporal domains (cardiac, respiratory, and T1/T2), datasets can be quite large and hard to properly train. Different groups have also experimented with directly estimating T1 (or T2) values from the DL network, rather than using the network to reconstruct images, followed by conventional T1 (or T2) fitting.

In a study by Chen et al., DL reconstructions were used to reconstruct large high-dimensional (5D) datasets containing 2D images with cardiac, respiratory, and inversion time dimensions [113]. To deal with the large data size and GPU memory limitations, the network operates on a lower-dimensional temporal feature space, using similar techniques as in low-rank CS models. A densely connected network was used in this work and compared to a U-Net architecture and a traditional CS-based reconstruction that was considered the reference standard. The proposed network showed very little bias with the CS reconstruction and limits of agreement in the range of 50 ms.

Jeelani et al. designed a reconstruction method called DeepT1 that contains a recurrent CNN for reconstructing images, which is then followed by a U-Net on the output images to obtain T1 maps, rather than traditional fitting [114]. This was tested using a 2D gated sequence with eight modified look-locker inversion recovery times, in pre- and post-contrast acquisitions. The DL methods were compared to fully sampled images, where good agreement was seen with the rCNN image reconstruction and even better agreement when using the UNet to generate T1 maps from the images (Figure 6).

MR fingerprinting is another technique used to measure T1 and T2 in the heart, which uses a series of highly undersampled images with pseudo-random flip angles to solve for relaxation times. Hamilton et al. designed a method for reconstructing T1 and T2 values from fingerprinting images to speed up the solving process [115]. They used a densely connected network with pixel-wise signal curves as input in order to output both T1 and T2. The method is trained from Bloch simulations and was tested against a normal dictionary-based solver. They found that they achieved similar results with both methods, while the DL method performed in less than a second, compared to minutes for the conventional solver.

## 6. Pitfalls and Future Outlook

DL-based reconstructions have demonstrated the capacity to accelerate MRI image acquisitions, either with higher acceleration rates or with higher accuracy than CS. DL-based MRI reconstructions have shown the potential to enable more comprehensive and detailed examinations. However, these techniques present several pitfalls and challenges that need to be considered prior to their widespread clinical adoption. We highlight the challenges related to robustness and generalizability due to instabilities and interpretability, clinical deployment such as performance gaps, downstream tasks, and increased computational costs.

### 6.1. Instabilities and Hallucinations

DL-based reconstruction networks are potentially subject to instabilities and hallucinations. Antun et al. [116] showed that adding small, almost undetectable perturbations in the acquired *k*-space could introduce severe artifacts in the reconstructed images. Furthermore, the authors showed that small structural changes in the image, such as those arising from tumors, may not be captured in the reconstructed image. Similarly, Darestani et al. [117] studied instabilities and hallucinations and showed that even though the reconstruction networks are prone to instabilities and hallucinations, more traditional methods such as $\ell_{1}$-norm minimization are sensitive to such problems as well. In fact, the authors showed that the reconstruction networks are actually more robust to these phenomena than traditional methods. Approaches to studying, identifying, and minimizing instabilities and hallucinations are a very active and critical area of research that will require collaboration between scientists and clinicians.

### 6.2. Interpretability

Instabilities and hallucinations are a major concern for deep learning methods, mainly because their inner workings are still not fully understood. While CS has a solid mathematical basis that assures robustness [31,32], the interpretability of deep neural networks is currently an active area of research. An interesting approach has used convex duality to assess shallow networks [118,119]. Additionally, Sahiner et al. [120] extended this framework to assess CNN-based image reconstruction networks. These works use convex duality to reformulate the non-convex training problem as convex in a higher-dimensional space. The alternative formulation offers convergence guarantees to the global optimum and, thus, improved robustness. Furthermore, this formulation provides further insight and interpretability. In the convex formulation, the authors found that weight decay regularization during training induced path sparsity regularization (i.e., the network enforces that the least possible number of filters are used). Additionally, once trained, the authors found that the network performs inference as a piece-wise linear filtering procedure. In other words, based on the input, the trained network will choose which filters to use (a non-linear procedure) and then perform conventional filtering (a linear procedure). Interpretability of reconstruction networks is an important area of research that should be given further attention. Understanding the inner workings of reconstruction networks could potentially provide insight into the cause of instabilities and hallucinations, and into possible ways to prevent them.

### 6.3. Performance Gaps

DL-based reconstruction methods are able to outperform conventional methods by learning tailored priors directly from data. Nevertheless, once deployed in the clinic, these methods can sometimes present a performance drop. Researchers have assessed this phenomena and reported two main reasons: (1) the mishandling of publicly available data and (2) distribution shifts in the data. Next, we elaborate on both topics.

Training datasets that are made publicly available are vital for continued DL-based reconstruction research. However, the growing availability of these public datasets may sometimes lead to “off-label” usage, where data published for one task are used for another. Shimron et al. [74] assessed two cases where data mishandling resulted in biased or overly optimistic results, namely when the *k*-space data have been zero-padded or when the data have been JPEG-compressed.

Likewise, DL-based methods are prone to performance drops due to distribution shifts. These methods are usually evaluated by the in-distribution performance. In other words, researchers divide the same dataset into training and testing subsets. Then, they use the training subset for training the network while they use the test subset to evaluate the performance. However, in clinical practice, performance is often evaluated in different distributions. For example, training and test datasets may come from different scanners, different sequences, or simply different noise levels [117]. All of these possible distribution shifts may introduce a performance drop of the reconstruction networks.

Darestani et al. [121] denoted this performance drop as a “distribution shift performance gap”. Furthermore, the authors proposed a domain adaptation method that successfully reduced the performance gap by 87%–99%. The method had two steps. First, during training, they incorporated a self-supervised loss to the conventional supervised loss. Second, during inference, they included extra self-supervised training, denoted as “test-time training”.

### 6.4. Downstream Tasks

Advanced DL-based reconstruction techniques that enable fast acquisitions and reconstructions will have significant clinical impact for the diagnosis and treatment of numerous cardiovascular diseases. However, these reconstruction methods should not overlook the downstream clinical processes. For example, in the case that the downstream task is a quantitative analysis such as in flow imaging, the need for speed should not come at the expense of compromised accuracy and precision in the estimation of the hemodynamic parameters of interest. Novel undersampling and DL-based reconstruction strategies should be developed without arbitrarily selecting the undersampling rate but by selecting the maximum achievable undersampling rate while still maintaining measurement accuracy and precision. Likewise, in the case of a downstream segmentation tasks, researchers should ensure compatibility with the desired segmentation method (i.e., if the DL-based reconstruction produces images that impact the segmentation process). Furthermore, some works [122,123] have developed DL methods that synergistically combine both reconstruction and segmentation tasks to improve both tasks.

### 6.5. Increased Computational Cost

A crucial component of deployment includes ensuring the users’ institutions have proper computation power to perform the reconstructions. While most DL-based reconstruction methods can outperform conventional reconstruction methods in terms of image quality and computation time, these methods oftentimes come at the expense of increased computational cost, requiring expensive GPUs for both training and inference. This limitation is a clinical bottleneck, especially for 3D non-Cartesian applications [124], such as 4D-Flow [77], Dynamic Contrast-Enhanced MRI (DCE) [48], and MR Fingerprinting [125]. Possible solutions to reduce the computation cost of these types of reconstructions include coil compression [126,127,128], stochastic gradient descent [48,129], and randomized sketching algorithms [130,131,132,133].

### 6.6. Improved Models and Losses

The field of machine learning moves incredibly fast, with new techniques that outperform the previous state of the art being published routinely. We expect an increase in achievable MRI data undersampling rates in the upcoming years due to novel and more sophisticated reconstruction methods. For example, the incorporation of a CNN-based perceptual loss can yield sharper edges and a more faithful image contrast as described by Wang et al. [134]. DL-based reconstruction performance depends heavily on the loss function used. However, commonly used losses such as $\ell_{1}$, $\ell_{2}$, or SSIM do not capture the perceptual information of fine structures, which results in reduced perceptual quality and blurring. The authors presented a method that yielded improved reconstructed image quality in knee datasets.

Additionally, transformer networks are a DL-based architecture with the potential to improve reconstruction in CMR. Since their introduction in 2017 [135], transformers have rapidly become the state of the art in natural language processing and have recently demonstrated the ability to outperform CNNs for image and video classification tasks [136,137]. These methods split the data into patches that are input to self-attention layers that extract features and are able to learn long-range dependencies. This architecture was already successfully applied to MR image reconstruction [138,139], and applications to CMR are expected soon.

## 7. Conclusions

Machine learning has disrupted the CMR field with novel reconstruction methods that have largely outperformed conventional methods both in image quality and reconstruction speed. DL-based reconstruction methods have the capacity to enable unprecedented fast acquisitions, providing a significant impact on the diagnosis and treatment of numerous diseases. For CMR, DL-based methods have been developed for virtually all applications, including cine, flow imaging, late gadolinium enhancement, and tissue characterization. However, there are still some pitfalls and limitations, such as the instabilities, interpretability, and increased computational cost. Finally, the machine learning field moves incredibly fast. New techniques that can potentially be applied to CMR image reconstruction are constantly being published. Thus, we expect that the pitfalls and challenges discussed will be overcome in the near future by novel DL-based reconstruction techniques, which is essential in order to fully translate these methods into the clinic.

## Figures and Tables

**Figure 1 bioengineering-10-00334-f001:**
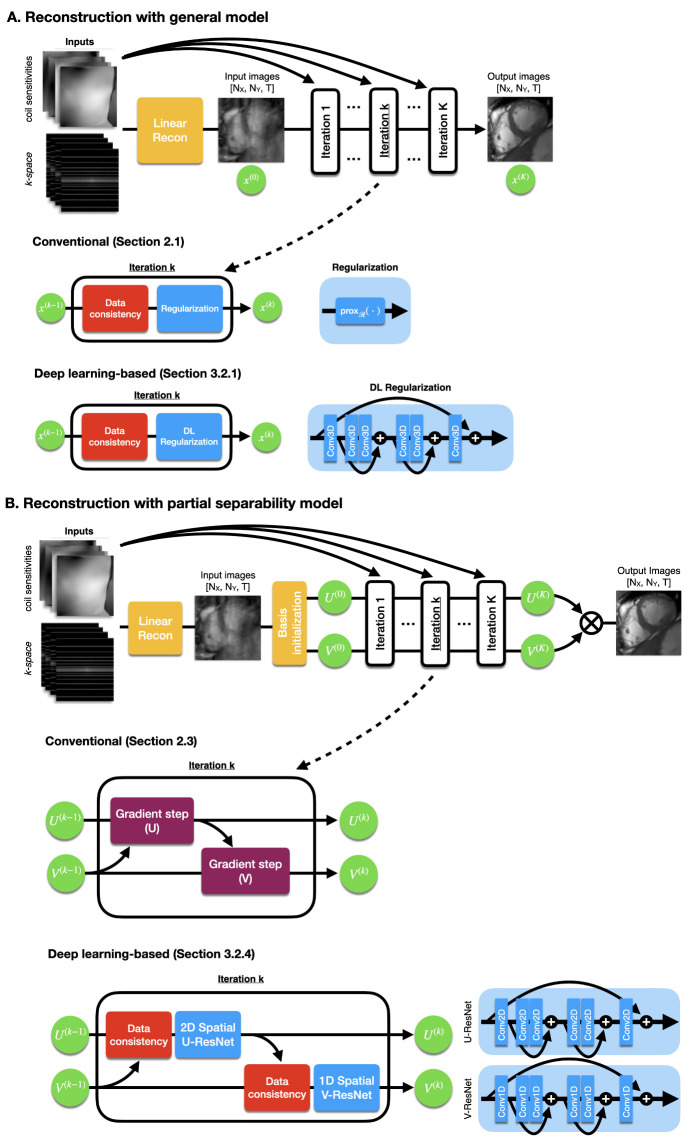
(**A**) A generalized image reconstruction algorithm iteratively removes aliasing artifacts produced by undersampling. The methods alternate between data consistency and regularization steps. A common conventional approach (e.g., [22]) would apply regularization via a proximal step. DL-based reconstruction methods regularize via a neural network. (**B**) An alternative reconstruction algorithm uses the partial separability model. Image reconstruction is performed on spatial and temporal basis rather than directly on the images. A conventional method (e.g., [47,48]) alternately updates the basis U and V via gradient steps. Similarly, a DL-based reconstruction method would update via spatial and temporal networks. Diagrams and pictures modified from Sandino et al. [49].

**Figure 2 bioengineering-10-00334-f002:**
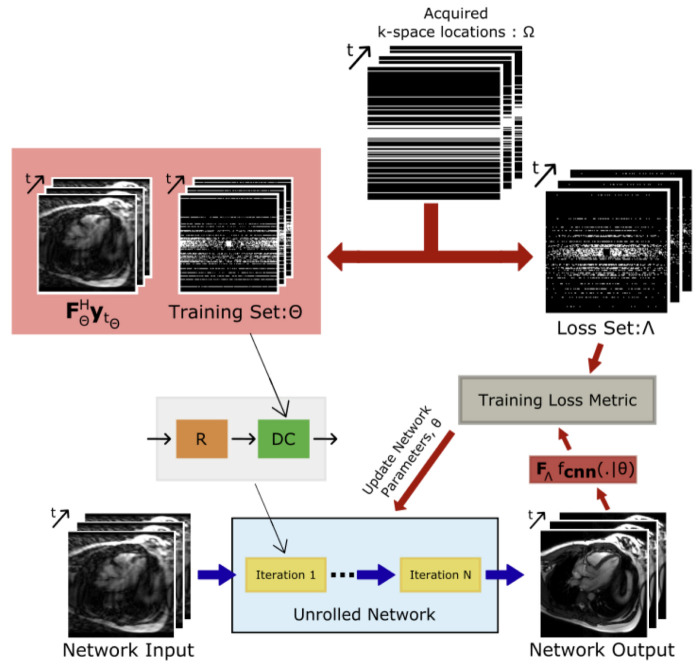
Self-supervised learning diagram for CMR image reconstruction from Acar et al. [46]. Acquired *k*-space Ω is split into two sets: training set Θ and loss set Λ. The training set Θ is used to enforce data consistency and the loss set Λ is used in the loss function to learn the network weights.

**Figure 3 bioengineering-10-00334-f003:**
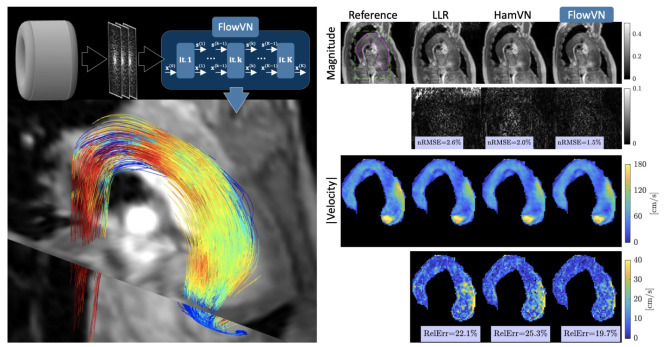
A representative retrospectively (R = 10) acquired dataset from Vishnevskiy et al. showing a patient with stenotic aortic flow at peak systole. A quantitative comparison of highlighting their FlowVN reconstruction comparing (nRMSE) and relative error (RelErr) for both magnitude and velocity images relative to a fully sampled dataset (reference) and a CS reconstruction.

**Figure 4 bioengineering-10-00334-f004:**
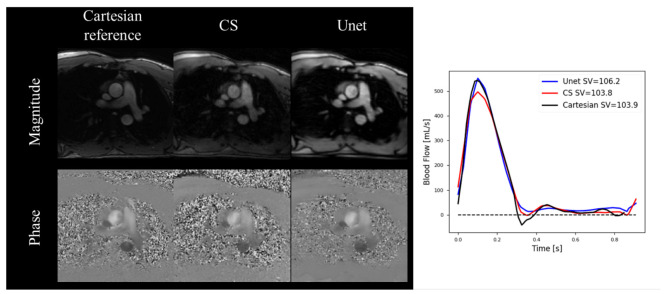
A representative prospectively (R = 18) acquired subject from Jaubert et al. showing the Cartesian reference as well as the CS and U-Net reconstruction at peak systole. Corresponding average real time–blood flow curves, averaged over 9-s, show more accurate peak blood flow rates for the U-Net compared to CS while also showing accurate stroke volumes for both CS and U-Net reconstructions.

**Figure 5 bioengineering-10-00334-f005:**
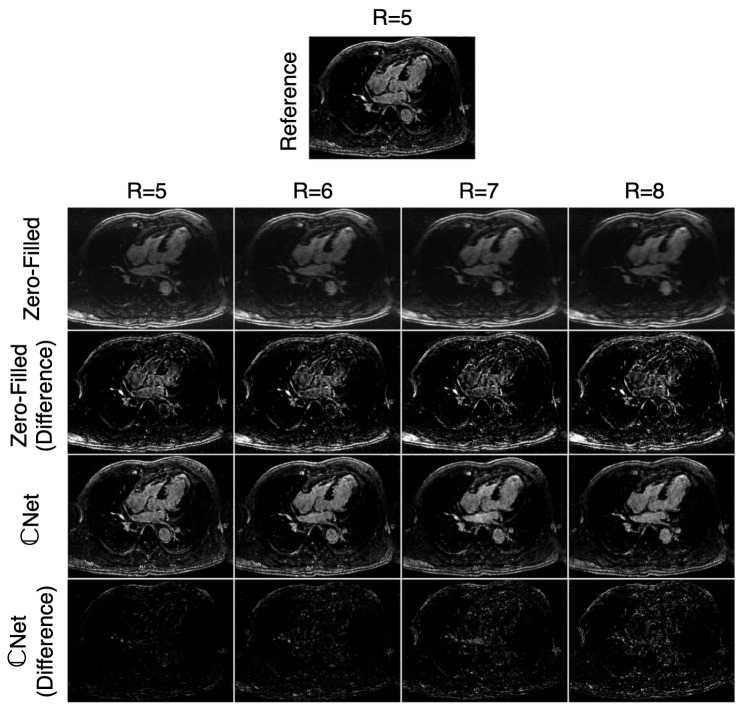
Representative LGE images from El-Rewaidy et al. acquired using prospective undersampling for R = 5 reconstructed with the reference method as well as retrospectively undersampled using R = 6, 7, and 8 reconstructed with zero-filling and the proposed DL-based reconstruction, CNet. Minimal noise and blurring artifacts are seen in the CNet images. Similarly, image quality is similar between R = 5 reference images and R = 8 CNet images while also maintaining scar–blood contrast.

**Figure 6 bioengineering-10-00334-f006:**
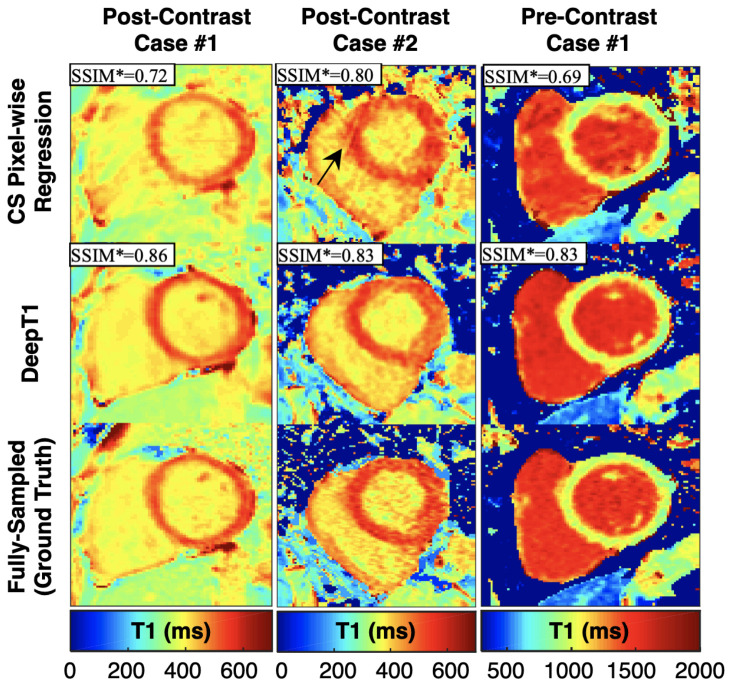
Representative T1 maps from Jeelani et al. for two post-contrast and one pre-contrast patient cases, comparing results between CS, DeepT1, and a fully sampled ground truth. DeepT1 shows better results than CS (higher SSIM compared to fully sampled).

**Table 1 bioengineering-10-00334-t001:** Overview of 2D- and 4D-Flow deep learning-based reconstructions.

Manuscript	Imaging	Network	Training	Undersampling	Recon Reduction
Vishnevskiy [97]	4D-Flow	Unrolled	n = 11	12.4 ≤ R ≤ 13.8	30×
Haji-Valizadeh [98]	2D-Flow	3D U-Net	n = 510	R = 28.8	4.6×
Cole [99]	2D-Flow	2D U-Net	n = 180	R ≤ 6	N/A
Jaubert [100,101]	2D-Flow	3D U-Net	n = 520	R = 18	15×
Oscanoa [39]	2D-Flow	2D U-Net	n = 155	R = 8	N/A
Kim [102]	4D-Flow	Unrolled	n = 140	R ≤ 6	N/A
Nath [103]	4D-Flow	2D U-Net	n = 18	R = 2.5, 3.3, 5	7×

Reconstruction time improvement is relative to compressed sensing (CS).

## Data Availability

Not applicable.

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
