# Peer review of "Deep Learning-Based Reconstruction for Cardiac MRI: A Review"

_bioengineering, 2023, doi:10.3390/bioengineering10030334_

Round 1
Reviewer 1 Report
The manuscript “Deep learning-based reconstruction for cardiac MRI: A review” describes the basics of deep learning for image reconstruction (with a comparison to compressed sensing), specific deep learning methods applied to CMR, CMR-specific challenges and applications, and open questions. This is a very well-written and comprehensive manuscript that will be an excellent guide to the entire MRI community.
Overall, I have very few comments; most are minor. The only major comment I have is that cine imaging seems to be treated in a very different way in this manuscript compared with other applications. In my experience, cine is the most commonly tackled acceleration problem in CMR (using deep learning and also more conventional approaches). Right now, cine is only discussed in the section which describes various deep learning structures for image reconstruction. It may be helpful to keep this format, and add a small paragraph (or even table) discussing all of the work which has been done in DL+cine to the “Application-Specific Methods” section. Another idea would be to specifically state that Section 3 “Deep Learning-based Reconstruction” is focusing on cine methods. Either way, calling out the fact that cine is a prime target for DL acceleration would be important, and I think that this gets lost in the current organization (in which cine is not mentioned in the “Application-Specifici Methods” section.
In a similar note, while the authors discuss the importance of training data, how the DL networks for accelerated cine scans (in Section 3) are trained is not mentioned at all. What sort of data are needed, and how much were used for each of these approaches? Comparing and contrasting the methods in terms of the training data required may be interesting to the reader.
Additional minor comments are listed below:
1) The words “herein” and “therein” are used several times in the beginning of the manuscript, which becomes a bit distracting. The authors may consider rephrasing
2) I found the labels on Figure 1, part A to be confusing. Which is approach falls under the conventional label and which is Deep Learning Based?
3) I don’t think that the acronym “VD” appears in Table 1.
4) Line 518, page 15 “Additionally, a batch normalization layer was used that better preserved the magnitude and phase relationships in the complex data. When compared to a standard CS reconstruction, the complex DL network performed similar 5, while reconstructing significantly (300x) faster. The network was also compared to a real-valued convolutional network which performed slightly worse.” Perhaps would be better to specify what in particular is improved instead of using terms like “similar” or “worse”.
5) The title of Section 4 is “Challenge-Specific Methods” which seems to be a funny name for this section; perhaps this could be altered to “Application-Specific Challenges”?
6) There are a few places where the wording could be adjusted, including:
Page 9, line 320: Due to the particular characteristics CMR
Page 17, line 590: perturbationss
Page 18, line 631: Namely, when the k-space data has been zero-padded or when the data has been JPEG-compressed (this is not a complete sentence)
Reviewer 2 Report
In this paper, the authors reviewed the DL-based reconstruction methods for cardiac magnetic resonance (CMR). They introduced the conventional reconstruction theory and connect them to several DL-based reconstruction methods. Then the challenge-specific methods and application-specific methods of CMR image reconstruction were presented. The authors also discussed the future work of the field in this review. The paper has a clear structure and good writing. Here are some suggestions to improve the paper:
- In figure 1 B, U and V seem to be obtained by SVD, but this constraint is not mentioned in the main text, please explain more about that.
- In Section 3, D_Φ is a denoising function performed by a deep neural network, which is used for regularization, as shown in Figure 1. I wonder how the regularization is performed and why denoising can be used instead of a proximal step.
- The authors mention the unrolled network many times in the paper. please give some details or an example structure of the unrolled network.
- I suggest that the authors add a sub-section to discuss the evaluation of DL-based reconstruction methods, which is very important for algorithm application.
- It is much better to connect the structure in Figure 1 with the Section number. So the readers can have a better understanding of Sections 2 and 3 by Figure 1.
Reviewer 3 Report
These present work could be a significant manuscript in the literature. It is well written and structured. Furthermore, has scientific value.
Author Response
The reviewer did not provide points to revise.
Reviewer 4 Report
The paper is well written and the topic is very interesting. Therefore, I suggest to accept the paper.
Author Response

(The authors gave the same response as above.)
